# Recognition of EEG Signals from Imagined Vowels Using Deep Learning Methods

**DOI:** 10.3390/s21196503

**Published:** 2021-09-29

**Authors:** Luis Carlos Sarmiento, Sergio Villamizar, Omar López, Ana Claros Collazos, Jhon Sarmiento, Jan Bacca Rodríguez

**Affiliations:** 1Departamento de Tecnología, Universidad Pedagógica Nacional, Bogotá 111321, Colombia; olopezv@pedagogica.edu.co (O.L.); asclarosc@upn.edu.co (A.C.C.); jfsarmientov@pedagogica.edu.co (J.S.); 2Department of Electrical and Electronics Engineering, School of Engineering, Universidad Nacional de Colombia, Bogotá 111321, Colombia; sivillamizard@unal.edu.co (S.V.); jbaccar@unal.edu.co (J.B.R.)

**Keywords:** imagined speech, electroencephalography, brain-computer interface (BCI), deep learning, convolutional neural networks (CNN), vowels

## Abstract

The use of imagined speech with electroencephalographic (EEG) signals is a promising field of brain-computer interfaces (BCI) that seeks communication between areas of the cerebral cortex related to language and devices or machines. However, the complexity of this brain process makes the analysis and classification of this type of signals a relevant topic of research. The goals of this study were: to develop a new algorithm based on Deep Learning (DL), referred to as CNNeeg1-1, to recognize EEG signals in imagined vowel tasks; to create an imagined speech database with 50 subjects specialized in imagined vowels from the Spanish language (/a/,/e/,/i/,/o/,/u/); and to contrast the performance of the CNNeeg1-1 algorithm with the DL Shallow CNN and EEGNet benchmark algorithms using an open access database (BD1) and the newly developed database (BD2). In this study, a mixed variance analysis of variance was conducted to assess the intra-subject and inter-subject training of the proposed algorithms. The results show that for intra-subject training analysis, the best performance among the Shallow CNN, EEGNet, and CNNeeg1-1 methods in classifying imagined vowels (/a/,/e/,/i/,/o/,/u/) was exhibited by CNNeeg1-1, with an accuracy of 65.62% for BD1 database and 85.66% for BD2 database.

## 1. Introduction

Brain-computer interfaces (BCI), also referred to as human-machine interfaces, are systems that use brain signals to control computers or hardware devices [1,2,3]. These systems can use invasive or noninvasive recording methods, where the latter stand out because they do not require surgical interventions [4]. Research on BCI is aimed at developing technological solutions in fields like motor and cognitive rehabilitation [5]; assistance in the recovery of compromised communication and/or physical skills [4]; control of video games [6]; augmentative assistance platforms [7,8,9], among others, aimed at improving the user’s quality of life and well-being.

Imagined speech (IS) is an innovative technique for BCI applications using voluntary signals. [10,11,12]. Imagined speech is the internal pronunciation of phonemes, words, or sentences, without the movement of the phonatory apparatus or any audible output [13]. In this sense, previous imagined speech works with conventional machine learning (ML) methods for imagined vowel recognition (/a/,/e/,/i/,/o/,/u/), have chosen to use time, frequency, or time-frequency transformations as the feature vector. Among the features that have been used for imagined vowel recognition (/a/,/e/,/i/,/o/,/u/) with ML are: statistical descriptors (average power, mean, variance, and standard deviation) [14]; common special patterns (CSPs) filtering, and adaptive collection (AC) [15]; Discrete Wavelet Transform (DWT) [16,17]; eigenvalues of the covariance matrix [18]; and mixed features such as descriptors (mean, variance, standard deviation, and skewness) and Sparse Regression Models [19].

Thus, to improve imagined speech signal recognition we propose to: increase the electrode density in the frontotemporal brain area of the left hemisphere according to Hickok and Poeppel’s model [20]; acquire EEG signals with cognitive imagined speech tasks under controlled artifact conditions (blinking and eye movement), environmental noise, and lighting; and implement several DL CNNs specialized in classifying pairs of patterns so they together can perform multiclass classifications. Accordingly, the objectives proposed for this work are: first, to develop a new DL based algorithm, referred to as CNNeeg1-1, for EEG signal recognition in imagined vowel tasks. This algorithm should perform a multiclass classification based on the specialization of a set of CNNs for the recognition of imagined vowel pairs. Second, to develop an imagined speech database with 50 subjects (BD2), under artifact-controlled conditions using electroencephalographic signals from the somatosensory areas in the left hemisphere of the brain, for Spanish imagined vowels (/a/,/e/,/i/,/o/,/u/). Finally, contrast the performance of the newly developed CNNeeg1-1 algorithm with the DP Shallow CNN and EEGNet reference algorithms for databases BD1 and BD2, with analysis of the intra-subject and inter-subject learning.

The remainder of this article is organized as follows: Section 2 is an overview of previous research in speech imagery with EEG for vowels; Section 3 presents the CNN algorithm developed in this research, as well as the materials and methods used; Section 4 describes the results obtained with the algorithms CNNeeg1-1, Shallow CCN, and EEGNet, Section 5 presents the discussion of the process and the results obtained in classifying vowel imagery speech signals (/a/,/e/,/i/,/o/,/u/). Finally, Section 6 presents the conclusions of this re-search.

## 2. Related Work

There are different methodologies for the non-invasive capturing of brain signals such as magnetoencephalography (MEG) [21,22], functional magnetic resonance imaging [23,24] and electroencephalography (EEG) [25,26]. The advantages that electroencephalography has over the other methods are its low cost, portability, and high time resolution [27]. The stages of a BCI processing system with EEG are: signal acquisition, preprocessing, feature extraction, classification, and device control [7].

Among the types of noninvasive EEG signals used for BCI control are evoked potentials and voluntary signals [28]. The first require external stimuli and include signals such as: Event Related Potential (ERP), Evoked Potential (P300), Movement Related Cortical Potential (MRCP), and Steady State Evoked Potentials (SSEP) [28]. On the other hand, voluntary signals are produced autonomously by the user such as: sensorimotor rhythms (SMR), slow cortical potentials (SCP), motor imagery (MI), and non-motor cognitive signals [28]. The studies conducted with MI sought to mimic motor intention (without using the muscular system), mainly using event-related desynchronization or event-related synchronization (ERS/ERD) signals [3,8]. However, MI requires a high degree of training to mitigate the effects of user attention and the consequent mental fatigue [4,9]. Within the context of voluntary signals, BCI systems are being developed based on high-level cognitive processes such as: mental mathematical operations, visual counting, musical imagination, imagined speech, among others [1,28]. One of the advantages of these new methods is the number of tasks that can be classified. However, these new methods are limited by the current knowledge in the field of neuroscience, cognitive science, artificial intelligence, among others.

Some classifiers that have been used for imagined vowel recognition (/a/,/e/,/i/,/o/,/u/) are summarized in the following table (Table 1):

In addition, it is noteworthy that the complexity in the processing of EEG signals is mainly due to: their voltage range (μV), their low signal to noise ratio (SNR), their non-linearity, non-temporality, and low spatial resolution given by the EEG electrodes. According to these characteristics, conventional ML methods are limited for the recognition of this type of signals [29,30]. This poses an important challenge in the design of new algorithms to identify the characteristics of the EEG signal [31,32] and, to select or design the proper classifiers [33,34]. In conclusion, an ideal method should be able to automatically recognize the inherent characteristics of the EEG signal with its nonlinear and nonstationary properties.

In consequence, a DL method has been proposed. DL is a subset of the field of ML, which learns input data representations through multiple layers of neural networks, whose architectures are based on human brain neural models [35,36,37]. Among the advantages of DL is the ability to automatically identify the characteristics of a signal [38,39].

Some DL architectures that have been used for imagined vowel recognition with EEG are summarized in the following table (Table 2):

Additionally, to reduce the effect of the low signal to noise ratio of EEG signals, there are alternative DL methods using EEG signal preprocessing for imagined vowels, such as: filtering from 2 Hz to 40 Hz, artifact detection and removal with Independent Component Analysis (ICA), and analysis with Hessian approximation preconditioning; eigenvalues of the covariance matrix [18]; 50 Hz LPF-IIR low-pass filters, 0.5 Hz HPF-IIR high-pass filters, and feature vectors consisting of EEG coherence, partial directed coherence (PDC), Direct Transfer Function (DFT) and transfer entropy [40].

Although, DL architectures have been successfully applied to image recognition [43,44,45] and speech signal recognition [46,47,48], their use for EEG signal recognition tasks, such as imagined speech [49,50], remains a challenge and requires the development of novel pre-processing techniques and the development of new DL structures and architectures [51,52]. Among the difficulties posed by DL algorithms are: CNN methods are susceptible to the effect of artifacts present in EEG signals, generating a reduction in the accuracy of the classifiers [27]. These methods are also affected by the reduced amount of data used in the training process [27]; the use of brain rhythms with fixed ranges as inputs to different CNNs can cause a decrease in classification accuracy since some of these rhythms may not provide the information needed for the system to extract the features from the targeted EEG signal [27]; DNN has accuracy limitations due to the number of subjects in the sample, inter-subject analysis, and the amount of time an experiment may take [28]; DL with Shallow CNN, Deep CNN and EEGNet are susceptible to the size of the experimental dataset and the reduced number of tests per class [17]. Additionally, DL architectures are susceptible to overfitting, which consists of overtraining the neural networks, generating a decrease in classification accuracy during testing [35]. The Shallow CNN and EEGNet architectures are going to be used as benchmarks for the proposed architecture, so they are described in more detail in Appendix A and Appendix B.

## 3. Materials and Methods

### 3.1. Data Description

This research used the reference database developed by Coretto et al. that involved 15 subjects and the imagined vowel tasks (/a/,/e/,/i/,/o/,/u/) [16]. It also includes a new database with 50 individuals, recorded under controlled conditions, for imagined vowels (/a/,/e/,/i/,/o/,/u/) developed specifically for this research.

#### 3.1.1. Reference Database (BD1)

The reference database (BD1) (http://fich.unl.edu.ar/sinc/downloads/imagined_speech/ accessed on: 24 September 2020) is an open-access database of EEG signals, developed by Coretto et al. which records imagined speech tasks with five vowels and five words [16]. For this article, we used the information of imagined vowels (/a/,/e/,/i/,/o/,/u/). This experiment was conducted with 15 Spanish-speaking higher education Argentine students (7 females and 8 males) between 24 and 28 years of age [16].

The experimental protocol for this database consisted in asking each subject to sit on a chair one meter away from an LCD screen. Once seated, they were shown a message on the screen for two seconds warning them to get ready. Then, they were shown the vowel they had to imagine for two seconds. Next, they imagined the vowel continuously for four seconds. Finally, they were shown a message on the screen indicating them to rest for four seconds. This procedure was repeated 40 times for each imagined vowel [16].

In this database, the signals were recorded with an 18-electrode Grass device at a sampling frequency of 1024 Hz. The EEG electrodes were located according to the international 10–20 system and the database contains information from six electrodes F3, F4, C3, C5, P3, and P4 [16].

#### 3.1.2. New Database (BD2)

This new database, created by us specifically for this study, held the information of 50 university students (20 women and 30 men) whose native language is Spanish (M = 24.76, SD = 7.66) (https://github.com/carlos-sarmientov/DATABASE-IMAGINED-VOWELS-1 accessed on: 4 August 2021). The participants did not exhibit any medical or neurological conditions. The experiment was approved by the Ethics Committee of the School of Medicine at Universidad Nacional de Colombia and the subjects gave written consent for their participation.

The experiment was conducted in the Cognition and Intelligent Systems laboratory at Universidad Pedagógica Nacional (Bogotá-Colombia), under controlled conditions: 80 lm/m^2^ lighting and minimum environmental noise (ASTM STC 63). First, each subject was asked to sit on a comfortable chair and an EEG neuroheadset was placed on their heads. The neuroheadset has 14 electrodes located on the left hemisphere, covering the language area. Two reference electrodes were located on the forehead. The electrodes were placed according to Hickok and Poeppel’s neurological model of language related to the sensorimotor interface and articulatory network (Broca’s area and motor cortex) related to Brodmann areas: 4, 6, 43, 44 and 45 [20]. The electrodes were placed on the neuroheadset in a matrix-like structure where the rows and columns of electrodes, were 18 mm apart. To reference the neuroheadset on the head of each subject, the T3 and C3 positions were used according to the 10–20 system (Figure 1). Once the headset was secured, a light source, placed at one meter from the subject, was lit to indicate the moment when they should start or finish the task of thinking about a specific vowel with imagined speech. To decrease blinking and eye movement artifacts, subjects were asked to keep their eyes closed.

For the experiment, each subject was told to imagine a given vowel continuously and without pronouncing it while the light source was on. They were also told that, when the light source was turned off, they had to stop imagining the vowel and relax their body. During the experiment, the light source remained on for four seconds and then was turned off for three seconds. The procedure was repeated 25 times for each one of the imagined vowels. Upon completion of the 25 imagined speech tasks for each vowel, subjects rested for 5 min to continue with the next vowel. The imagined tasks were arranged in the following order: /a/,/e/,/i/,/o/,/u/(Figure 2).

The signals were recorded with a 14-channel EMOTIV EPOC+ amplifier, with a sampling frequency of 128 Hz, a 14 bits resolution with 1 LSB with 0.51 μV in monopolar configuration. The 14 electrodes of the EMOTIV EPOC+ device were arranged on the neuroheadset (E1, …, E14) paying attention to the original name of each electrode of the device. For this experiment, the electrodes are numbered from E1 to E14 and the relationship with the original name of the Emotiv electrodes is as follows: E1 (AF3), E2 (F7), E3 (F3), E4 (FC5), E5 (T7), E6 (P7), E7 (O1), E8 (O2), E9 (P8), E10 (T8), E11 (FC6), E12 (F4), E13 (F8), E14 (AF4). The two reference electrodes were placed on the subject’s forehead (Figure 1).

The EpocSimulinkImporter acquisition software from Xcessity (Linz, Austria) was used to export the data to Matlab’s Simulink. Signal preprocessing was performed with Matlab R2020a. Additionally, signal processing was performed with: Matlab R2020a using the Deep Learning Toolbox for the CNNeeg1-1 model, Jupyter Notebook (Anaconda3) with Python 3.0 using TensorFlow and Keras for the Shallow CNN and EEGNet models. Data analysis was carried out using the Statistical Package for the Social Sciences (SPSS) Version 25 software (Armonk, NY, USA).

### 3.2. Deep Learning Methods with Convolutional Neural Networks (CNN)

Following is the description of this research’s proposed architectures. The first one corresponds to the new proposed method. Another two benchmark methods using CNN reported for imagined speech are included [17].

#### 3.2.1. CNNeeg1-1 Architecture

The proposed architecture consists of 10 signal preprocessing blocks for each one of the 10 CNNs, used for the recognition of imaged vowel pairs and one stage for the one-against-one function (1-1) that allows multi-class classification of imagined vowels (/a/,/e/,/i/,/o/ and /u/). The proposed DL-based architecture is described below.

##### Preprocessing

The proposed architecture consists of 10 preprocessing blocks that filter and adapt the brain signals to deliver it to each CNN. Each preprocessing block is mainly composed of a filtering stage using Adaptive-Projection Intrinsically Transformed MEMD (APIT-MEMD) and a signal transformation stage using spectral analysis. The brain signals recorded were edited to keep only the intervals in which the subjects performed the corresponding imagined speech tasks. The signals were divided in trials with 64 samples and an overlap of 85%.

For the filtering process, the APIT-MEMD method was chosen since the signals have nonlinear and nonstationary characteristics [53]. This method separates the multivariate signals into so-called Intrinsic Mode Functions (IMFs). It includes the following steps [53]:For each multidimensional input frame ${\left[x\left(t\right)\right]}_{t=1}^{T}$ and each shift operation x(t), decompose the covariance matrix as $C=E\left\{ss^{T}\right\}=W\Lambda W^{T},$ where $W=\left[w_{1},\;w_{2},\;\ldots ,\;w_{n}\right]$ is the eigenvector matrix, and $\Lambda =diag\left\{\lambda_{1},\lambda_{2},\;\ldots ,\;\lambda_{n}\right\}$ is the eigenvalue matrix. In this case the largest eigenvalue will correspond to the eigenvector $w_{1}$.Take the first principal component and build a vector pointing in the opposite direction to $w_{01}=-w_{1}$.Using the Hammerseley sequence on a uniformly sampled sphere, build a set of K direction vectors ${\left\{p^{\theta_{k}}\right\}}_{k=1}^{K}$.Calculate the Euclidean distances from each of the uniform direction vectors to $w_{1}$.Relocate half of the projection vectors $p_{w_{1}}^{\theta_{k}}$, the closest to $v_{1},$ using ${\hat{p}}_{w_{1}}^{\theta_{k}}$=$\frac{{\hat{x}}_{w_{1}}^{\theta_{k}}+\alpha w_{1}}{\left|{\hat{x}}_{w_{1}}^{\theta_{k}}+\alpha w_{1}\right|}$, where α is used to control the density of the relocated vectors.The other half of the uniform projection vectors, ${\hat{p}}_{w_{1}}^{\theta_{k}}$, the closest to $w_{01}$, are relocated using ${\hat{p}}_{w_{01}}^{\theta_{k}}$=$\frac{{\hat{x}}_{w_{01}}^{\theta_{k}}+\alpha w_{10}}{\left|{\hat{x}}_{w_{01}}^{\theta_{k}}+\alpha w_{01}\right|}$, where α is used to control the density of the relocated vectors.Project the multidimensional signal ${\left[x\left(t\right)\right]}_{t=1}^{T}$ along the direction vectors found in steps 5 and 6.Find the instant of time $t_{i}^{\theta_{j}}$ corresponding to the maximum of the projected data sets, where $\theta_{j}$ is the angle of the $\left(n-1\right)$ dimensional sphere and j is the index of the direction vectors.Interpolate $\left[t_{i}^{\theta_{j}}\;\left(x_{i}^{\theta_{j}}\;\right)\right]$ to calculate the envelope curves ${\left[e^{\theta_{j}}\left(t\right)\right]}_{j=1}^{J}$.Estimate the mean of the envelope curves for the set of direction vectors J:$$m=\frac{1}{J}\overset{J}{\underset{j=1}{\sum }}e^{\theta_{j}}\left(t\right)$$Calculate the residue $d\left(t\right)=x\left(t\right)-m\left(t\right)$.Repeat these steps until the residue meets the conditions of an IMF for multivariate signals.

The first two IMFs resulting from applying the APIT-MEMD algorithm to the brain signals, (IMF1, IMF2), are chosen for this architecture. They have center frequencies of approximately 30 Hz and 15 Hz, respectively (Figure 3). These two IMFs are added for each one of the 14 electrodes.

With the signals obtained from APIT-MEMD, a transformation between electrode pairs is performed according to the following equation: $abs\left(FFT\left(E_{i}\right)-FFT\left(E_{j}\right)\right)$, where $E_{i}$ and $E_{j}$ represent each electrode, where i,j=1,…,14, and j>i. The values are normalized between 0 and 1. After this, each trial of databases BD1 and BD2 is converted into a jpeg-image. The images are 15 × 32 for BD1 and 91 × 32 for BD2. The rows of these images correspond to the frequencies and the columns correspond to the pairs-differences between electrodes. Database BD1 results in 1888 images for each imagined vowel, for a total of 9440 images for the training and testing of the CNNs. Database BD2 produces 1274 images for each imagined vowel, for a total of 6370 images for the training and testing of the CNNs.

##### CNNeeg1-1: A New Deep Learning Architecture with CNN

The proposed architecture consists of 10 CNNs using deep learning for one of the imagined speech pairs: (/a/-/e/), (/a/-/i/), (/a/-/o/), (/a/-/u/), (/e/-/i/), (/e/-/o/), (/e/-/u/), (/i/-/o/), (/i/-/u/), (/o/-/u/) (Figure 4).

A NVIDIA GeForce GTX 1080 Ti GPU with 11 Gbps next generation GDDR5X memory and a large frame buffer of 11 GB was used. The algorithm was implemented with Matlab 2020a using the Deep Learning Toolbox. For the training of the CNN networks, the stochastic gradient descent with momentum (SGDM) optimizer was used. The learning rate chosen was 0.01. The number of epochs was 50. In this way, the values of hyperparameters learning rate, training epochs and activation function were selected according to [17]. 70% of the data was used for training and 30% for validation. The architecture of the CNNs is described below.

The input layer for each CNN receives the information of the images obtained from the EEG imagined vowels. It consists of a tensor of size 32 × 15 × 1 for database BD1 and 32 × 91 × 1 for database BD2 (Table 3). Next comes the dropout layer that randomly sets, for each input image, a mask with 25% of its elements to zero, with the goal of minimizing the overfitting in the training process (Table 3). Layer 3 is a 2D convolutional layer that applies a sliding convolution filter on the input. For this layer, 50 filters are configured with a size of 5 × 5, a stride of 1 × 1, and a padding of 0; thus, the output has a size of 28 × 57 × 50 (Table 3). In layer 4, a batch normalization is applied to improve the training of the convolutional networks and reduce the sensitivity to network initialization. It is applied to the 50 input channels of the layer. In layer 5, the reluLayer function is applied, where
(1)$$f\left(x\right)=\left\{\begin{array}{c} x,\;x\geq 0\\ 0,\;x<0 \end{array}\right.$$

Layer 6 is a max pooling layer where a downsampling divides the input into rectangular regions. Then, the maximum value of each region is calculated (Table 3). The size of each region was 2 × 2, with a stride of 2 × 2, and a padding of 0; thus, the output has a size of 14 × 43 × 50 (Table 3).

Next, a 2D convolutional layer is implemented in layer 7. For this layer, 50 filters with size of 11 × 11, a stride of 1 × 1, and a padding of 0 are configured; thus, the output has a size of 4 × 33 × 60. In layer 8 a batch normalization is applied to the 50 input channels of the layer, and then, in layer 9, the reluLayer function is applied. Layer 10 corresponds to a max pooling layer where the size of each region was selected as 2 × 2, with a stride of 2 × 2, and a padding of 0; thus, the output has a size of 2 × 16 × 60. In layer 11, a batch normalization is applied to improve the training of the convolutional networks and reduce the sensitivity to network initialization, in this case applied to the 60 channels of the previous layer (Table 3).

In layers 12 and 13, two fully connected layers are implemented, multiplying the inputs by a weight matrix to which the corresponding bias vector is added (Figure 5). Layer 12 has an output size of 60 and layer 13 has an output size of 2, corresponding to the number of classes of each one of the 10 CNNs. Subsequently, in layer 14 (Table 3), the softmax function that calculates cross entropy loss for the corresponding classes is applied. Finally, layer 15 corresponds to the classification output layer of the corresponding CNN.

Then, the classification information of the 10 CNNs, is fed to a last block called one-against-one (1-1) [54]. The one-against-one function (1-1) has 10 inputs corresponding to the binary classifier outputs of the 10 CNNs (Figure 4). With a voting scheme, the predictions made by CNN (/a/-/e/), CNN (/a/-/i/), CNN (/a/-/o/), CNN (/a/-/u/), CNN (/e/-/i/), CNN (/e/-/o/), CNN (/e/-/u/), CNN (/i/-/o/), CNN (/i/-/u/), CNN (/o/-/u/) are combined, and the class receiving the largest share of the vote is the winner within the imagined vowels (/a/,/e/,/i/,/o/,/u/) [54], and it is chosen as the output of this last block.

It is important to underline that CNNeeg1-1 is composed by ten CNN-type algorithms designed to extract the characteristics of the magnitude difference of the FFT of the EEG signals obtained through silent speech. Such differences were calculated between pairs of electrodes. Each CNN of CNNeeg1-1 is based on machine vision architectures with DL [43,44] and classic CNN architectures like LetNet5 and AlexNet [35], since their effectiveness has already been shown. Speaking of the actual architecture, the first layer of each CNN of CNNeeg1-1 is a Dropout layer whose goal is to apply to the image a mask with a certain percentage of ceros randomly located. This layer intends to diminish the possible overfitting resulting from the training of each CNN. The next two blocks contain four layers each as follows: 2D-convolution, Batch normalization, Non-linearity, and Max-Pooling. The goal of the first block is for each CNN to learn the characteristics of the frequency signals through their convolution with 50 × 5 × 5 spatial filters. The goal of the second block is for each CNN to learn the characteristics of the outputs of the first block. This process is performed through convolution with 50 × 11 × 11 spatial filters. The parameters used in these two blocks were obtained through a swept of a value grid, looking to maximize accuracy. With the characteristics found in the training process, the algorithm moves on to the classification stage, made up of two Fully Connected layers and a Softmax layer. The first Fully connected layer is made up of 60 neurons and the second one of 2 since it must classify two types of silent speech signals. The results obtained with the CNNeeg1-1 architecture proposed were compared with the Shallow CNN and EEGNet architectures. These are described in Appendix A and Appendix B respectively.

## 4. Results

### 4.1. Analysis of Intra-Subject Training Results for the Shallow CNN, EEGNet, and CNNeeg1-1 Algorithms Using Databases BD1 and BD2

The intra-subject training process consists in taking the brain signals from silent speech tasks of each one of the subjects independently, disregarding the ones from the other subjects. The set of signals from each subject is split randomly in a training set, with 70% of the signals, and a testing set, with 30% of the signals. This process is repeated for each subject in each database independently. In consequence, the information of both databases is kept apart, they do not mix.

The statistical analysis, for intra-subject training process, was done using a variance mixed analysis of repeated measures. In this case, using BD1 database, the following results were obtained: For Shallow CNN, a mean and standard deviation accuracy of (M = 0.3171, SD = 0.0114) was achieved. EEGNet achieved an accuracy of (M = 0.3506, SD = 0.0133). Finally, CNNeeg1-1 obtained an accuracy of (M = 0.6562, SD = 0.0123) (Figure 5).

For the BD2 database, the results were: For Shallow CNN, an accuracy of (M = 0.5371, SD = 0.0606) was achieved. EEGNet obtained an accuracy of (M = 0.7068, SD = 0.0396). Finally, CNNeeg1-1 had an accuracy of (M = 0.8566, SD = 0.0446) (Figure 6).

Mauchly’s test indicated that the assumption of sphericity was violated (X(2) = 46.546, *p* < 0.05), therefore, the degrees of freedom were adjusted with Greenhouse-Geisser (ε = 0.654). Tests for intra-subject effects show significant differences between the classification of imagined vowels performed by the three CNN models with F (1.31,82.46) = 1017.50, *p* < 0.001, η2 = 0.942. Similarly, the results show that there is a significant interaction between the intra-subject (CNN Model) and inter-subject (database) variable related to the accuracy F(1.31,82.46) = 64.40, *p* < 0.001, η2 = 0.506.

According to the inter-subject analysis, related to database type, there is a significant difference between database BD1 (M = 0.441, SD = 0.008) and database BD2 (M = 0.700, SD = 0.005) in imagined vowels recognition with F(1,63) = 738.12, *p* < 0.001, η2 = 0.921 (Figure 7).

Below we discuss the post-hoc analysis, according to Bonferroni, highlighting the significant differences between pairs of variables in imagined vowel classification processes in terms of accuracy. Starting with BD1 database, there are significant differences between the Shallow CNN model (M = 0.3171, SD = 0.0114) and the EEGNet model (M = 0.3506, SD = 0.0133) (*p* < 0.05) (Figure 5). Also, there are significant differences between the Shallow CNN model (M = 0.3171, SD = 0.0114) and the CNNeeg1-1 model (M = 0.6562, SD = 0.0123) (*p* < 0.05). There are also significant differences between the EEGNet model (M = 0.3506, SD = 0.0133) and the CNNeeg1-1 model (M = 0.6562, SD = 0.0123) (*p* < 0.05) (Figure 7). Thus, in this comparison the CNNeeg1-1 model (M = 0.6562) has the highest mean followed by the EEGNet model (M = 0.3506).

Moving on to BD2 database related to imagined vowel classification, there are significant differences between the Shallow CNN model (M = 0.5371, SD = 0.0606) and the EEGNet model (M = 0.7068, SD = 0.0396) (*p* < 0.05) (Figure 6). Additionally, there are significant differences between the Shallow CNN model (M = 0.5371, SD = 0.0606) and the CNNeeg1-1 model (M = 0.8566, SD = 0.0446) (*p* < 0.05). Also, there are significant differences between the EEGNet model (M = 0.7068, SD = 0.0396) and the CNNeeg1-1 model (M = 0.8566, SD = 0.0446) (*p* < 0.05) (Figure 7). Thus, in this comparison the CNNeeg1-1 model (M = 0.8566) has the highest mean, followed by the EEGNet model (M = 0.7068).

There are also significant differences between the Shallow CNN model with BD1 database (M = 0.3171, SD = 0.0114) and BD2 database (M = 0.5371, SD = 0.0606) for imagined vowel recognition (*p* < 0.05). Also, there are significant differences in the EEGNet model relative to BD1 database (M = 0.3506, SD = 0.0133) and BD2 database (M = 0.7068, SD = 0.0396) for imagined vowel classification in terms of accuracy (*p* < 0.05). Additionally, there are significant differences of the CNNeeg1-1 model in one case contrasting database BD1 (M = 0.6562, SD = 0.0123) and in the other case contrasting database BD2 (M = 0.8566, SD = 0.0446) for recognition, in terms of imagined vowel accuracy (*p* < 0.05). Thus, for the three CNN models, the corresponding means for BD2 database are superior when compared to the CNN models for BD1 database (Figure 7).

### 4.2. Subject’s Internal Visualization BD2 Database CNNeeg1-1

To visualize the internal representation of the CNNeeg1-1 network, the CAM (Class Activation Mapping) method that predicts the network behavior using class activation was used [55]. The following figures show the internal visualization for a subject in imagined vowel tasks (/a/,/e/,/i/,/o/,/u/) using database BD2 in the layer BN_3 (Table 3). Each figure represents, on the horizontal axis, the pair-wise differences for the 14 electrodes from E1-E2 to E13-E14 and on the vertical axis, the corresponding frequencies. The colors represent the CAM value for each electrode pair and each frequency, which oscillates between 0 to 255.

Figure 8 shows a subject’s internal representation (CAM) for the task of imagining the vowel /a/. Some of the electrode pairs that are activated the most are: E1–E7 in the frequencies from 12 to 56 Hz; E3–E14 in the range from 56 to 60 Hz; E4–E12 ranging from 14 to 18 Hz; E7–E12, from 6 to 14 Hz and from 46 to 48 Hz; E9–E11 from 4 to 6 Hz; and E9–E12 from 4 to 6 Hz, from 32 to 38 Hz, and from 58 to 62 Hz.

Figure 9 shows a subject’s internal representation (CAM) for the task of imagining the vowel /e/. The electrode pairs that are activated the most are: E1–E2, between 36 to 38 Hz and 52 to 58 Hz; E5–E6, between 46 to 48 Hz; E5–E13, between 26 to 28 Hz; E9–E10, 44 to 46 Hz.

In the case of Figure 10, the subject’s internal representation (CAM) for the task of imagining the vowel /i/ is shown. For this case, the electrode pairs that are activated the most are: E1–E2, between 6 to 14 Hz; E1–E11 and E1–E12, between 18 to 22 Hz; E5–E7 and E5–E8, between 2 to 8 Hz; E7–E13, between a frequency of 26 to 30 Hz.

Figure 11 shows a subject’s internal representation (CAM) for the task of imagining the vowel /o/. In this case, the electrode pairs that are activated the most are: E1–E6 for frequencies between 30 to 32 Hz; E2–E10, between 24 to 30 Hz and 44 to 50 Hz; E4–E11, between 52 to 54 Hz; E7–E12, between 14 to 20 Hz and 52 to 62 Hz; E7–E14, between 34 to 3 Hz.

Figure 12 shows a subject’s internal representation (CAM) for the task of imagining the vowel /u/. The electrode pairs that are activated the most are: E2–E10 for frequencies between 24 to 28 Hz, 38 to 40 Hz, and 52 to 54 Hz; E2–E11 between 54 to 58 Hz; E4–E10 and E4–E11, between 14 to 18 Hz; E7–E12, between 8 to 12 Hz and 52 to 58 Hz; E7–E13, between 8 to 12 Hz and 32 to 34 Hz; E7–E14 between 32 to 34 Hz.

### 4.3. Analysis of the Inter-Subject Training Results for the Shallow CNN, EEGNet, and CNNeeg1-1 Algorithms Using BD1 and BD2 Databases

In contrast, the inter-subject training process takes the signals of all subjects in one of the databases used (15 subjects for database BD1 and 50 subjects for database BD2). When one of the CNN is trained for, for example subject 1 in BD1, the training set is defined as the data from the other 14 subjects in database BD1, except for subject 1, and the testing set is defined as the data from subject 1. For the actual training of the CNN, 70% of the training set is chosen randomly. Once the training is finished, the results are tested with 30% of the testing set, again chosen randomly. This process is then repeated for each one of the remaining subjects in database BD1. The same process is carried on with database BD2 independently, that is, the information in both databases is not combined.

The statistical analysis, for inter-subject training, was conducted using repeated measures mixed ANOVA. For the inter-subject training in the case of BD1 database, Shallow CNN obtained a mean and standard deviation of (M = 0.2587, SD = 0.0157) in accuracy. In the case of EEGNet, an accuracy of (M = 0.3531, SD = 0.0277) was achieved. Finally, CNNeeg1-1 presented an accuracy of (M = 0.5008, SD = 0.0133) (Figure 13).

For the inter-subject training with BD2 database, Shallow CNN achieved (M = 0.2475, SD = 0.0245). EEGNet has an accuracy of (M = 0.4578, SD = 0.0433). Finally, CNNeeg1-1 presents an accuracy of (M = 0.6276, SD = 0.0645) (Figure 14).

Mauchly’s test indicated that the assumption of sphericity was not met (X(2) = 29.749, *p* < 0.05), therefore, the degrees of freedom were adjusted with Greenhouse-Geisser (ε = 0.724). Tests for intra-subject effects show significant differences between the classification performed by the three CNN models for imagined speech of the vowels with F (1,448,91,231) = 1299.262, *p* < 0.001, η2 = 0.954. Similarly, the results show that there is a significant interaction between the intra-subject (CNN Model) and inter-subject (database) variable related to the accuracy F(1,448,91,231) = 73.723, *p* < 0.001, η2 = 0.539.

In the inter-subject analysis, related to database type, there is a significant difference between database BD1 (M = 0.371, SD = 0.009) and database BD2 (M = 0.444, SD = 0.005) in imagined vowel recognition tasks with F(1,63) = 50.377, *p* < 0.001, η2 = 0.444 (Figure 15).

The post-hoc analysis for the inter-subject training is discussed below, according to Bonferroni, highlighting the significant differences between pairs of variables in imagined vowel classification (accuracy) processes. Analyzing first the results obtained with BD1 database, significant differences were found between the Shallow CNN model (M = 0.2587, SD = 0.0157) and the EEGNet model (M = 0.3531, SD = 0.0277) (*p* < 0.05) (Figure 13). Also, there are significant differences between the Shallow CNN model (M = 0.2587, SD = 0.0157) and the CNNeeg1-1 model (M = 0.5008, SD = 0.0133) (*p* < 0.05). There are also significant differences between the EEGNet model (M = 0.3531, SD = 0.0277) and the CNNeeg1-1 model (M = 0.5008, SD = 0.0133) in terms of accuracy (*p* < 0.05). Thus, in this comparison the CNNeeg1-1 model (M = 0.5008) has the highest mean, followed by the EEGNet model (M = 0.3531) (Figure 15).

Moving on to BD2 database, there are significant differences between the Shallow CNN model (M = 0.2475, SD = 0.0245) and the EEGNet model (M = 0.4578, SD = 0.0433) (*p* < 0.05) (Figure 14). Additionally, there are significant differences between the Shallow CNN model (M = 0.2475, SD = 0.0245) and the CNNeeg1-1 model (M = 0.6276, SD = 0.0645) (*p* < 0.05). Also, there are significant differences between the EEGNet model (M = 0.4578, SD = 0.0433) and the CNNeeg1-1 model (M = 0.6276, SD = 0.0645) in terms of accuracy (*p* < 0.05) (Figure 15). Thus, in this comparison, the CNNeeg1-1 model (M = 0.6276) has the highest mean, followed by the EEGNet model (M = 0.4578).

To complement, for inter-subject training there are no significant differences for the Shallow CNN model with BD1 database (M = 0.2587, SD = 0.0157) or with BD2 database (M = 0.2475, SD = 0.0245) (Figure 15). However, there are significant differences for the EEGNet model with BD1 database (M = 0.3531, SD = 0.0277) and BD2 database (M = 0.4578, SD = 0.0433) in terms of accuracy (*p* < 0.05). Additionally, there are significant differences from the CNNeeg1-1 using database BD1 (M = 0.5008, SD = 0.0133) and using database BD2 (M = 0.6276, SD = 0.0645, in terms of accuracy (*p* < 0.05) (Figure 15). Thus, for the EEGNet and CNNeeg1-1 models, the corresponding means of BD2 database (M = 0.4577 AND M = 0.6276) are superior when contrasted with the CNN models for BD1 database (M = 0.3531 AND M = 0.5008). For the case of the inter-subject training of the Shallow CNN, BD1 database has a higher mean (M = 0.2587) than BD2 database (M = 0.2475), but the difference is not significant (Figure 15).

The Table 4 and Table 5 show the results of the training processes for the three CNN algorithms (Shallow CNN, EEGNet, and CNNeeg1-1) with database BD1 (Table 4) and database BD2 (Table 5). The tables specify the intra-subject and inter-subject training models in terms of accuracy.

## 5. Discussion

This research developed a new algorithm based on Deep Learning, referred to as CNNeeg1-1, designed for the recognition of imagined speech patterns (/a/,/e/,/i/,/o/,/u/) based on EEG signals (Figure 4). In addition, a new imagined speech database with 50 Spanish-speaking subjects, named BD2 was created. This database was recorded under artifact-controlled conditions. It is made up of electroencephalographic signals obtained according to Hickok and Poeppel’s speech production model, [20] involving the dorsal pathway between the sensorimotor interface and the articulatory network over the left hemisphere, in imagined vowel tasks (/a/,/e/,/i/,/o/,/u/). Finally, the performance of the CNNeeg1-1 algorithm was compared with two reference algorithms: Shallow CNN and EEGNet, performing an analysis of the intra-subject (Figure 7) and inter-subject (Figure 15) training process using database BD2 (50 subjects) and database BD1 (15 subjects), using a mixed variance analysis of repeated measurements.

The intra-subject training model CNNeeg1-1 with BD1 database (M = 0.6562, SD = 0.0123) (Figure 7) and database BD2 (M = 0.8566, SD = 0. 0446) (Figure 15) in EEG imagined vowel recognition (/a/,/e/,/i/,/o/,/u/) had an accuracy comparable or superior to other works developed with DL for imagined vowel recognition (/a/,/e/,/i/,/o/,/u/) such as: DBN with an accuracy of 80% with 6 subjects [18,40] and an accuracy of 87. 96% with 3 subjects [18]; with RNN an accuracy of 70% with 6 subjects [40]; with CNN an accuracy of 32.75% with 15 subjects [41,42] and an accuracy of 35.68% with 15 subjects [42]. In the case of Shallow CNN, Deep CNN, EEGNet, for 15 subjects, accuracies of 29.62%, 29.06%, and 30.08% respectively, have been achieved [17]. Thus, it is evidenced that the CNNeeg1-1 model has a better performance for the recognition of imagined vowels (/a/,/e/,/i/,/o/,/u/) compared to other DL methods.

On the other hand, studies developed with conventional techniques using machine learning in imagined vowel classification tasks (/a/,/e/,/i/,/o/,/u/) exhibit outstanding performance with algorithms such as: ELM, ELM-L, ELM-R, SVM-R, and LDA with accuracies from 50% to 90% with 5 subjects and 64 electrodes [19]; SVM-G, RVM-G, and RVM-L with accuracies from 77% to 79% with 5 subjects and 19 electrodes [15]; and with SVM, Random forest, rLDA with accuracies of 22.23%, 23.08%, and 25.82%, respectively, have been achieved with 15 subjects and 6 electrodes [17]. Thus, it is evident that the CNNeeg1-1 model (Figure 7 and Figure 15) has an accuracy that is comparable and in some cases higher for the recognition of imagined vowels (/a/,/e/,/i/,/o/,/u/) compared to the previously described ML methods.

The class activation mapping (CAM) method was used to observe the internal behavior of the CNNeeg1-1 algorithm (Figure 8, Figure 9, Figure 10, Figure 11 and Figure 12). In this study, it was possible to determine, for a given subject, the electrodes and frequencies that are activated more intensely in imagined vowel tasks (/a/,/e/,/i/,/o/,/u/). Thus, it is possible to determine that, for a specific subject, some of the areas that registered the highest activity were: for the imagined vowel /a/, the differences between electrodes E1–E7, E3–E14, E4–E12, E7–E12, E9–E11, and E9–E12 (Figure 8); for the imagined/e/, the differences between electrodes E1–E2, E5–E6, E5–E13, E9–E10 (Figure 9); for imagined/i/, the differences between electrodes E1–E2, E1–E11, E1–E12, E5–E7, E5–E8, E7–E13 (Figure 10); for the imagined/o/, the differences between electrodes E1–E6, E2–E10, E4–E11, E7–E12, E7–E14 (Figure 11); and for the imagined/u/, the differences between electrodes E2–E10, E2–E11, E4–E10, E4–E11, E7–E12, E7–E13, E7–E14 (Figure 12). This indicates that there are different electrodes, at different frequencies, located between the sensorimotor interface and the articulatory network related to the language areas of the Hickok & Poeppel model, involved in the tasks of imagined vowels (/a/,/e/,/i/,/o/,/u/) [20].

As a strategy to evaluate the imagined vowel (/a/,/e/,/i/,/o/,/u/) classification ability of the CNNeeg1-1 architecture (Figure 4), it was compared with two previously reported reference architectures for imagined vowels classification: Shallow CNN (Appendix A, Figure A1) and EEGNet (Figure A2). The results of intra-subject training with BD1 and BD2 databases indicate that there are significant differences between the three CNN models (Shallow CNN, EEGNet, and CNNeeg1-1) with F (1.31,82.46) = 1017.50, *p* < 0.001, η2 = 0.942. For the two databases in the intra-subjects training with post-hoc analysis it was found that there are significant differences between the models for each of the corresponding pairs (*p* < 0.05). This comparison evidenced that, for the case of BD1 database, the CNNeeg1-1 model obtained the highest average value (M = 0.6562, SD = 0.0123) (Figure 7). Similarly, for BD2 database, the CNNeeg1-1 model obtained the highest average value (M = 0.8566, SD = 0.0446) (Figure 7). The CNNeeg1-1 model not only recognizes imagined vowels (/a/,/e/,/i/,/o/,/u/), but also performs better by showing a higher accuracy than the Shallow CNN and EEGNet models.

For the case of inter-subject training, the results with databases BD1 (Figure 13) and BD2 (Figure 14) indicate significant differences between the three CNN models (Shallow CNN, EEGNet, and CNNeeg1-1) with F (1,448,91,231) = 1299.262, *p* < 0.001, η2 = 0.954. Post-hoc analysis showed significant differences between the models for each corresponding pair (*p* < 0.05). The comparison shows that, for BD1 database, the CNNeeg1-1 model obtained the highest average value (M = 0.5008, SD = 0.0133, *p* < 0.05). Similarly, for the case of BD2 database the CNNeeg1-1 model obtained the highest average value (M = 0.6276, SD = 0.0645, *p* < 0.05). Thus, it is evident that the CNNeeg1-1 model recognizes imagined vowels (/a/,/e/,/i/,/o/,/u/) with both databases in the inter-subject modality with a better performance than the Shallow CNN and EEGNet architectures (Figure 15).

When comparing BD1 database and BD2 database, regarding the intra-subject training process, for the three CNN models (Shallow CNN, EEGNet, and CNNeeg1-1) there are significant differences between both databases with F (1,63) = 738.12, *p* < 0.001, η2 = 0.921. For all three cases, the mean of each of the CNN architectures reported superior performance for BD2 database compared to BD1 database (*p* < 0.05) (Figure 7). In the case of the inter-subject training process, we found that, for the EEGNet and CNNeeg1-1 models, there are significant differences between BD1 database and BD2 database F (1,63) = 50.377, *p* < 0.001, η2 = 0.444, highlighting that the means are higher for BD2 database than for BD1 database (*p* < 0.05) (Figure 15). Thus, the performance of the CNNeeg1-1 architecture is verified by the results in the classification of imagined vowels for both BD1 and BD2 databases. Additionally, the number of subjects in each database: 15 subjects for BD1 database and 50 subjects for BD2 database, verifies the robustness of the CNNeeg1-1 algorithm.

For this study, we sought to place the electrodes (Figure 1) taking into account the speech production model of Hickok & Poeppel [20]. In this model, the speech production process is related to the dorsal branch of the sensorimotor interface and the articulatory network in the left hemisphere, the motor cortex and Broca’s area [20]. Thus, the available electrodes (14) were located aiming to cover the corresponding area of the cerebral cortex for the recording of BD2 database (Figure 1). In contrast, the recording of BD1 database was done placing three electrodes on the left hemisphere (F3, C3, and P3) around to the language area and three electrodes (F4, C4, and P4) on the right hemisphere [16]. There are significant differences between the results obtained with both databases with the three CNN models in the case of intra-subject training F (1,63) = 738.12, *p* < 0.001, η2 = 0.921 and for the EEGNet and CNNeeg1-1 models in the case of inter-subject training F (1,63) = 50. 377, *p* <0.001, η2 = 0.444. For these cases it was found that the accuracy values in the classification of imagined vowels is higher for BD2 database than for BD1 database (Figure 7 and Figure 15). This indicates that the placement of the electrodes covering the sensorimotor interface and the articulatory network of the Hickok and Poeppel [20] model contributes to the recognition of imagined vowels.

Comparing the BD2 database and BD1 database, we found that BD2 presents a higher accuracy (Figure 7 and Figure 15). One explanation for the higher performance of BD2 is given by the controlled characteristics of the experiment such as: controlled lighting conditions of 80 lm/m^2^ and controlled environmental noise conditions (ASTM STC 63). During the recording, the 50 participants were asked to remain seated without moving their limbs, this is reflected in the decrease of artifacts due to EMG type signals. Finally, during the acquisition of the signals, the subjects were asked to keep their eyes closed, in order to reduce artifacts generated by blinking and eye movement.

Regarding the preprocessing of the EEG signals, there are several DL methods that do not perform preprocessing of imagined vowel signals before they are delivered to the different DL architectures, but the results show generally low accuracy values in the classification of imagined vowels [17,41]. Other DL methods perform this preprocessing in different ways, such as: 2 Hz to 40 Hz filtering, artifact detection and removal with ICA and analysis with Hessian approximation preconditioning [42]; eigenvalues of the covariance matrix [18]; 50 Hz LPF- IIR low-pass filters and HPF-IIR high-pass filters of 0. 5 Hz and feature vectors with EEG coherence, partial directed coherence (PDC), direct transfer function (DFT), transfer entropy [40], among others. EEG signals have a low signal to noise ratio, and they are nonlinear and non-stationary. In this study, we chose to perform a preprocessing stage using APIT-MEMD and selecting just a few IMFs. This step is followed by the application of differences in the FFT of EEG signals between electrodes for the Shallow CNN, EEGNet, and CNNeeg1-1 models (Figure 4, Figure A1 and Figure A2).

Among the DL architectures that have been used for imagined vowel recognition are: DBN [18,40], RNN [40], CNN [41,42], Shallow CNN [17], and EEGNet [17]. All these architectures have tended to use a single neural network with different layers for multiclass recognition of the imagined vowels. Theses architectures have common elements such as: 2D convolution layers, max pooling layers, nonlinearity function layers, batch norm layers, etc. For this study, an architecture called CNNeeg1-1 was designed, which consists of 10 CNNs and a one-against-one fusion (Figure 4). Each CNN specializes in recognizing an imagined speech vowel pair: CNN (/a/-/e/), CNN (/a/-/i/), CNN (/a/-/o/), CNN (/a/-/u/), CNN (/e/-/i/), CNN (/e/-/o/), CNN (/e/-/u/), CNN (/i/-/o/), CNN (/i/-/u/), CNN (/o/-/u/) (Figure 5). According to the information received from the 10 CNNs, the 1-1 function selects with the one-against-one method the imagined vowel class (Figure 4). In this sense, the performance of the CNNeeg1-1 architecture is corroborated with the results in the classification of imagined vowels for both BD1 and BD2 databases (Figure 7 and Figure 15). Additionally, the number of subjects, 15 for BD1 database and 50 for BD2 database, verifies the robustness of the CNNeeg1-1 algorithm.

Among the limitations of the present study are the following: the capture and processing of brain signal data with imagined speech was performed offline. The experiment was carried out in a single session and it is advisable to perform several sessions in future research. It is advisable to increase the number of electrodes on the language area of the left hemisphere in future research. In the present study, we worked with imagined vowels, but we suggest exploring other language elements such as words. Finally, it is advisable to design other DL architectures to increase the accuracy in data classification.

In general terms, the method based on CNNeeg1-1 for imagined vowels classification does not require demanding training processes, as in the case of imagined motor tasks [56]; it does not require a rigorous attention process like in SSVEP [57,58], P300, or imagined motor tasks [59]; it does not require an external stimulus like SSVEP or P300 [60,61]; and it does not require cognitive tasks that generate muscular or cognitive fatigue as in imagined motor tasks [56,59]. Consequently, the CNNeeg1-1 method developed in this study has the potential to use other language components and to be applied in such relevant fields as BCI device control.

## 6. Conclusions

This study developed and tested a new algorithm called CNNeeg1-1 based on DL for EEG imagined vowel signal recognition using two different databases: BD1, with 15 subjects and BD2, with 50 subjects. The latter was created as part of the study. Among the factors that influenced the performance of CNNeeg1-1 are: the preprocessing stage based on the selection of IMFs calculated with the APIT-MEMD algorithm, together with the selection of the difference of the FFTs between electrodes; in the case of BD2 database, the location of the electrodes over the sensorimotor interface area and articulatory network of the left hemisphere based on the Hickok & Poeppel model; the proprietary architecture of the CNNeeg1-1 that uses 10 CNNs specialized in the recognition of imagined vowel pairs, feeding a one-against-one block, among others.

Additionally, the performance of the CNNeeg1-1 algorithm was compared with two reference algorithms with DL: Shallow CNN and EEGNet using both databases. Statistical results were presented with a mixed analysis of variance of repeated measures for intra-subject and inter-subject training. The results show that CNNeeg1-1 outperforms both Shallow CNN and EEGNet for EEG imagined vowel classification in intra-subject and inter-subject training analysis with both databases. Thus, it is shown that it is possible to classify imagined vowel with the new CNNeeg1-1 algorithm.

## Figures and Tables

**Figure 1 sensors-21-06503-f001:**
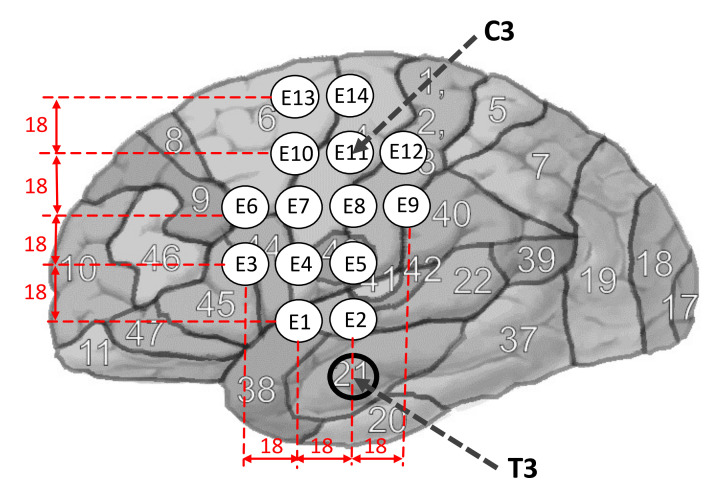
Location of the neuroheadset, which contains 14 electrodes (E1,…,E14) covering a section of the left hemisphere (language area). This is, the sensorimotor interface area and articulatory network of Hickok and Poeppel (Broca’s area and motor cortex) related to Brodmann areas: 4, 6, 43, 44 and 45. C3 and T3 are reference points from the 10–20 positioning system.

**Figure 2 sensors-21-06503-f002:**
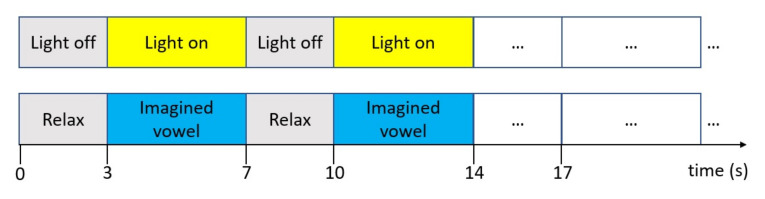
Time intervals for the imagined vowels experiment. When the light is on, the subject imagines the vowel and when the light is off, the subject relaxes.

**Figure 3 sensors-21-06503-f003:**
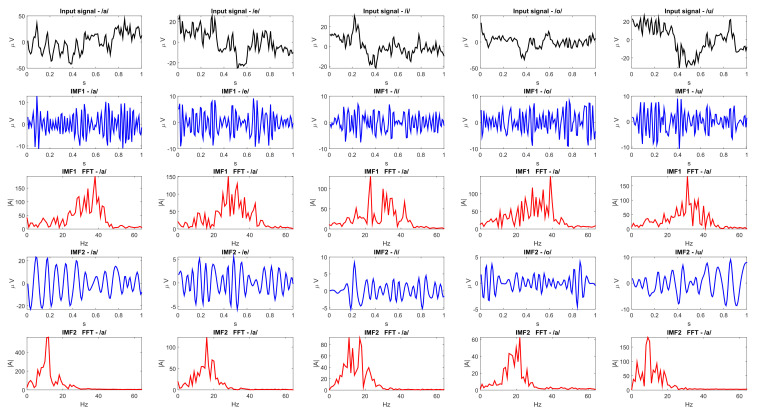
Application of the APIT-MEMD algorithm to a trial of imagined vowel signals (/a/,/e/,/i/,/o/,/u/), where the first two IMFs (IMF1 and IMF2) are shown in the time-domain (blue) and the frequency-domain (red) for a subject in BD2.

**Figure 4 sensors-21-06503-f004:**
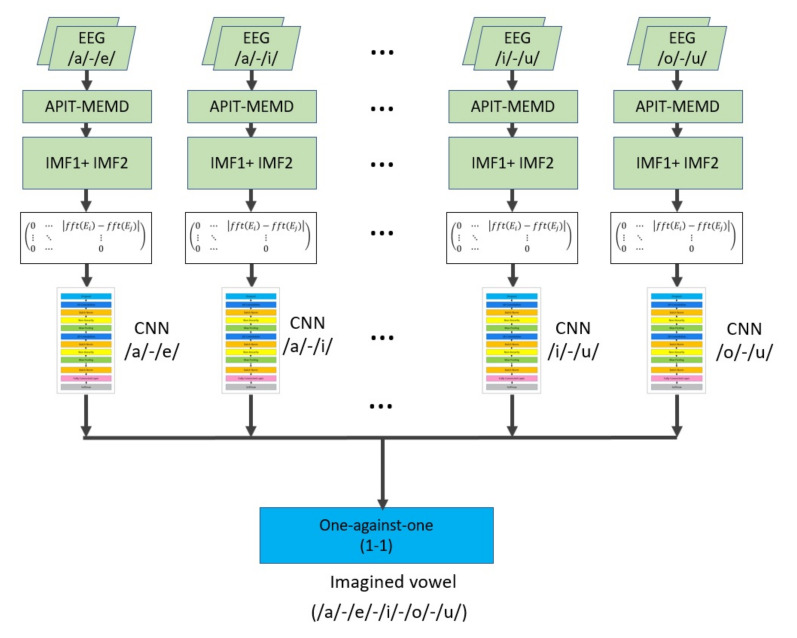
CNNeeg1-1 architecture made up of 10 specialized CNNs and a one-against-one function.

**Figure 5 sensors-21-06503-f005:**
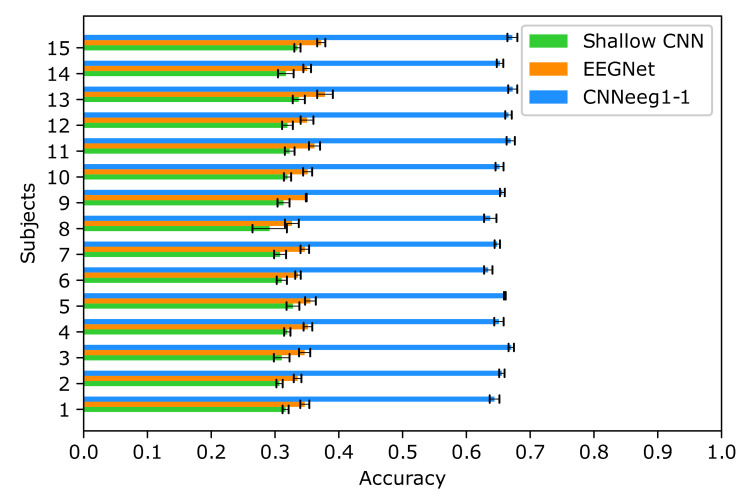
Intra-subject training classification accuracy for the Shallow CNN, EEGNet, and CNNeeg1-1 algorithms using BD1 database.

**Figure 6 sensors-21-06503-f006:**
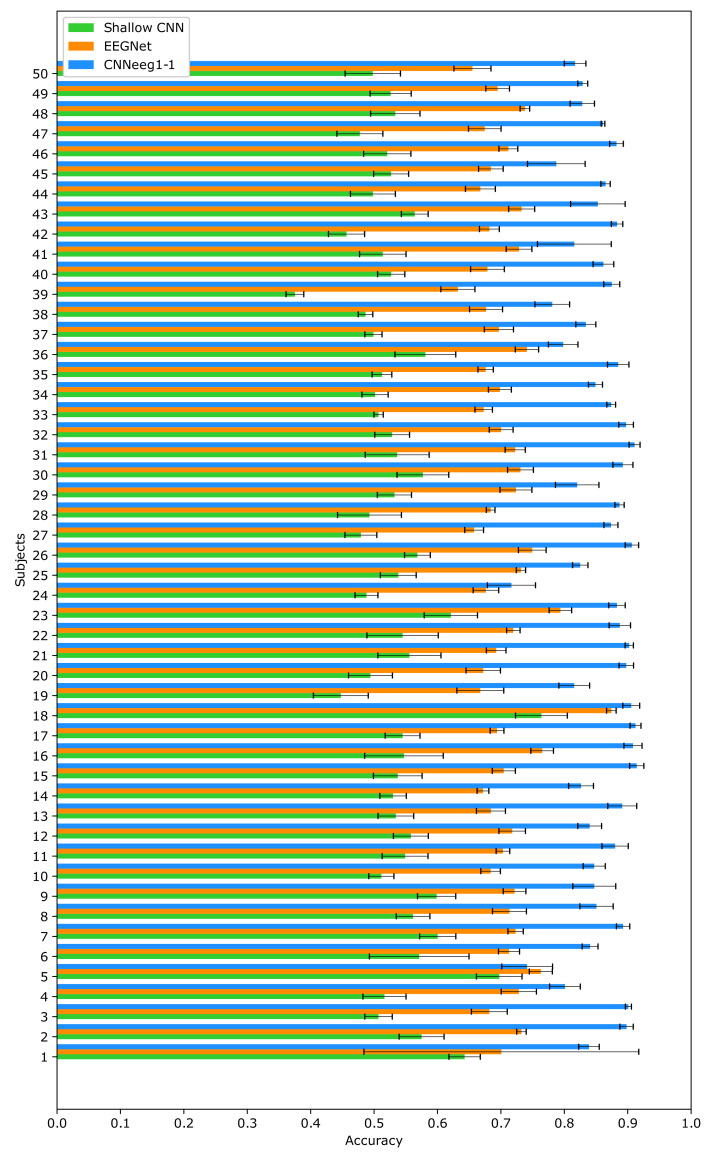
Intra-subject training classification accuracy for the Shallow CNN, EEGNet, and CNNeeg1-1 algorithms using BD2 database.

**Figure 7 sensors-21-06503-f007:**
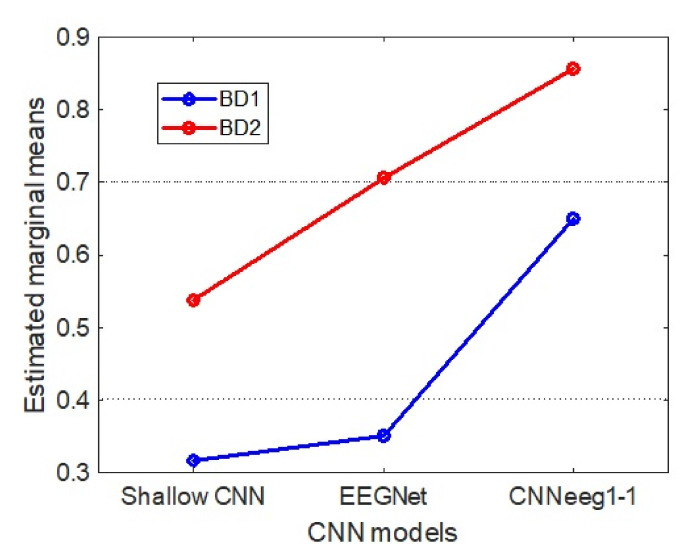
Marginal measures in intra-subject training classification for Shallow CNN, EEGNet, and CNNeeg1-1 algorithms using BD1 and BD2 databases.

**Figure 8 sensors-21-06503-f008:**
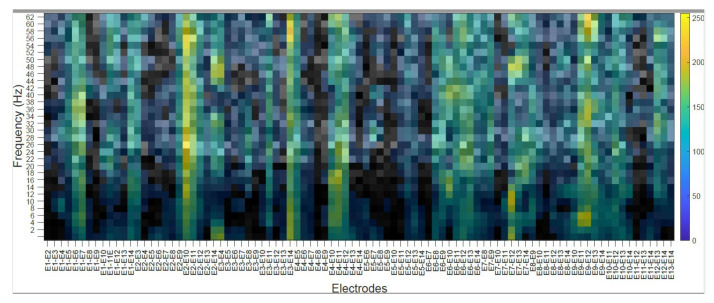
Subject’s Internal representation (CAM) in imagined tasks of the vowel /a/. The horizontal axis represents the difference between electrode and the vertical axis, the corresponding frequencies.

**Figure 9 sensors-21-06503-f009:**
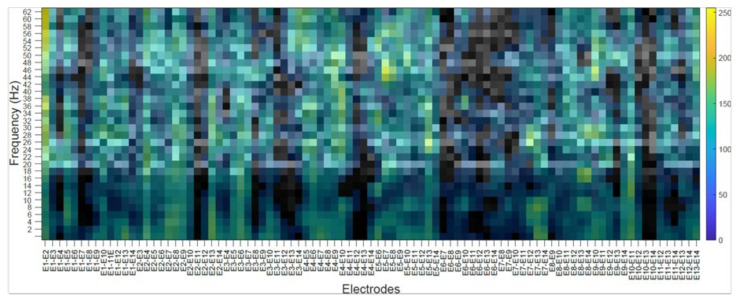
Subject’s internal representation (CAM) in imagined tasks of the vowel /e/. The horizontal axis represents, the difference between electrodes and the vertical axis, the corresponding frequencies.

**Figure 10 sensors-21-06503-f010:**
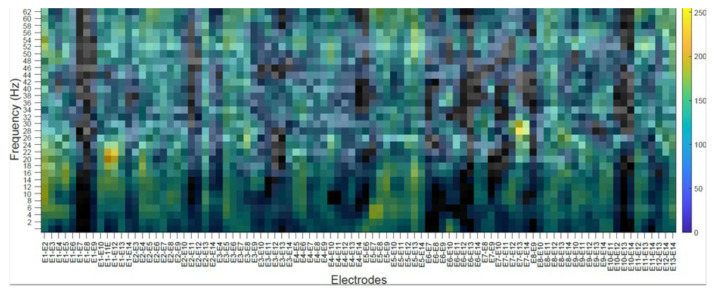
Subject’s internal representation (CAM) in imagined tasks of the vowel /i/. The horizontal axis represents the difference between electrodes and the vertical axis, the corresponding frequencies.

**Figure 11 sensors-21-06503-f011:**
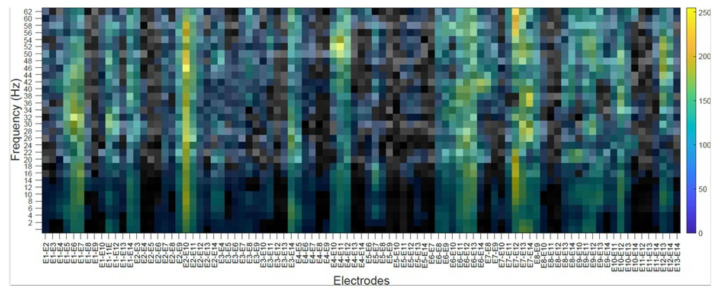
Subject’s internal representation (CAM) in imagined tasks of the vowel /o/. The horizontal axis represents the difference between electrodes and the vertical axis, the corresponding frequencies.

**Figure 12 sensors-21-06503-f012:**
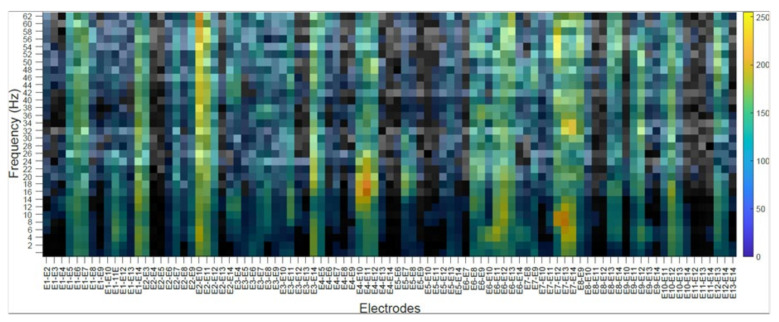
Subjects’ internal representation (CAM) in imagined tasks of the vowel /u/. The horizontal axis represents the difference between electrodes and the vertical axis, the corresponding frequencies.

**Figure 13 sensors-21-06503-f013:**
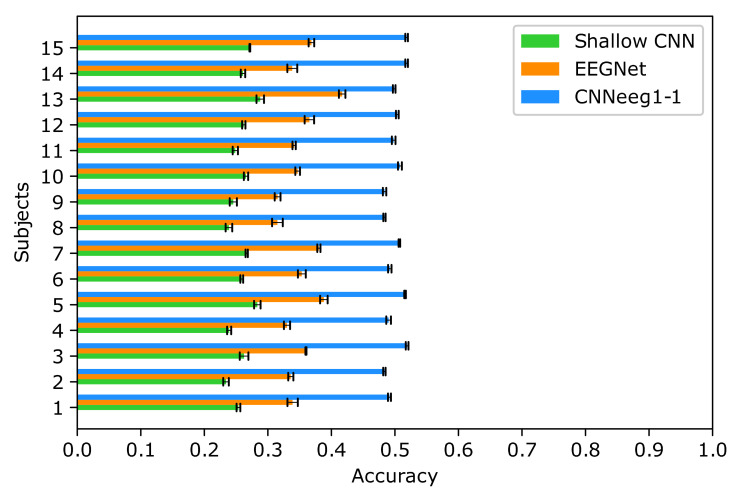
Inter-subject training classification accuracy for the Shallow CNN, EEGNet, and CNNeeg1-1 algorithms using BD1 database.

**Figure 14 sensors-21-06503-f014:**
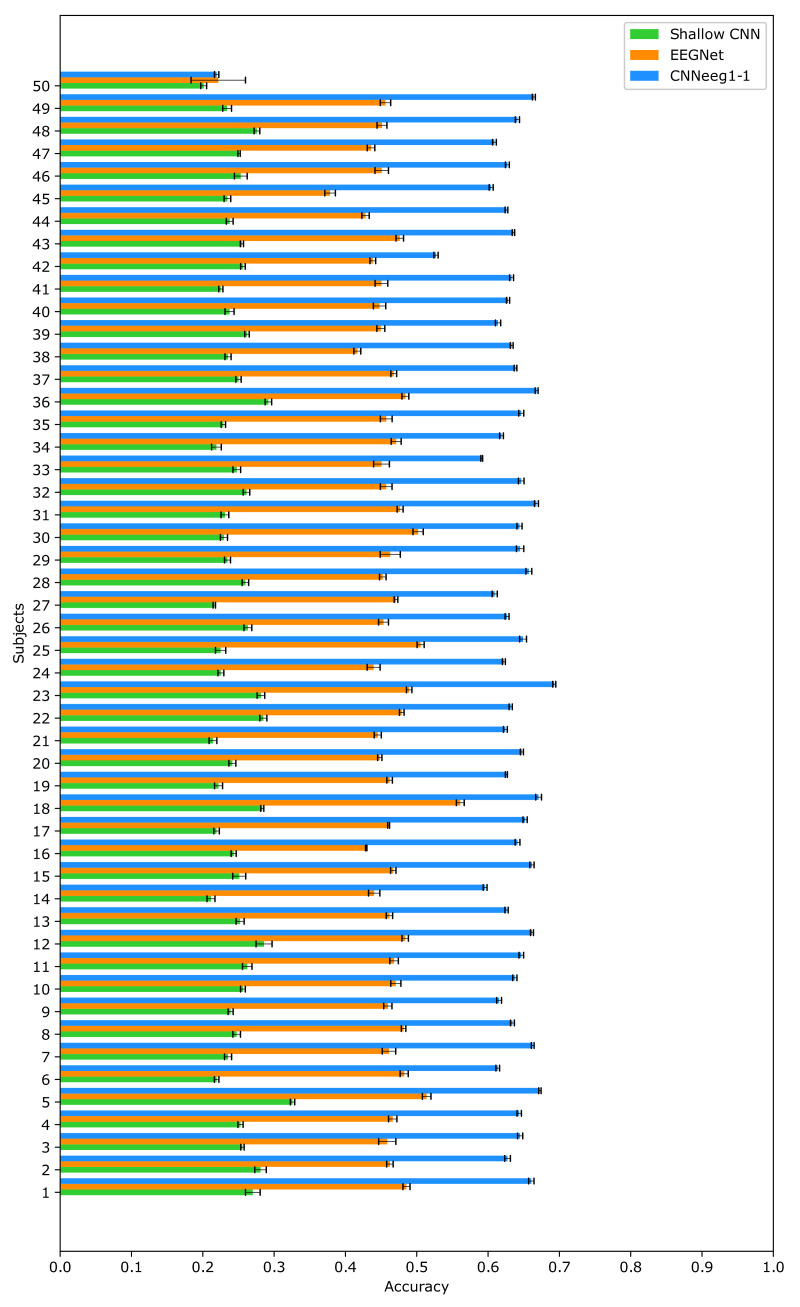
Inter-subject training classification accuracy for the Shallow CNN, EEGNet, and CNNeeg1-1 algorithms using BD2 database.

**Figure 15 sensors-21-06503-f015:**
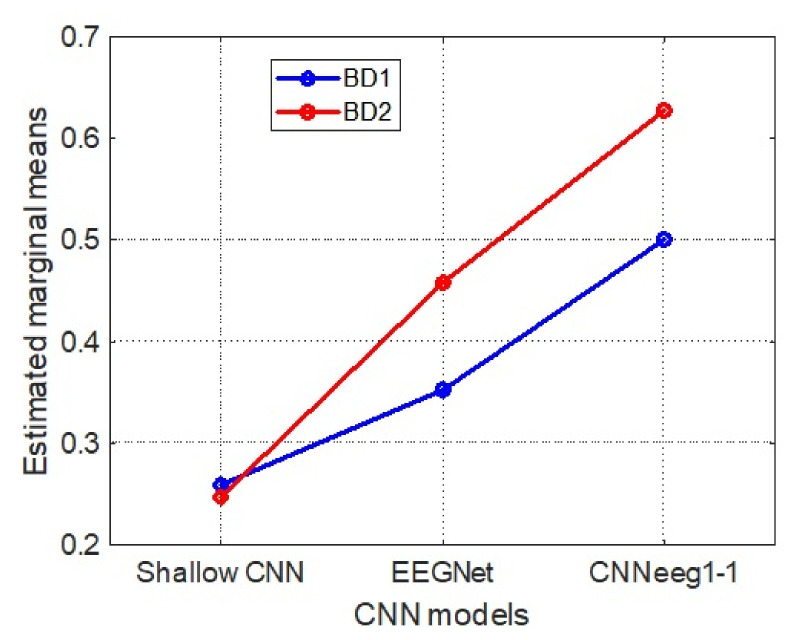
Marginal measures in the inter-subject training classification for the Shallow CNN, EEGNet, and CNNeeg1-1 algorithms using BD1 and BD2 databases.

**Table 1 sensors-21-06503-t001:** Machine learning classifiers used for imagined vowel recognition (/a/,/e/,/i/,/o/,/u/).

Classifiers	Accuracy	Subjects	Electrodes
Support Vector Machine with Gaussian kernel (SVM-G) [15]	77%	5	19
Relevance Vector Machine with Gaussian kernel (RVM-G) [15]	79%	5	19
Linear Relevance Vector Machine (RVM-L) [15]	50%	5	19
Bipolar Neural Network [14]	44%	13	19
Support Vector Machine (SVM) [16]	21.94%	15	6
Random Forest (RF) [16]	22.72%	15	6
Extreme Learning Machine (ELM) [19]	57–82%	5	64
Extreme Learning Machine with Linear Function (ELM-L) [19]	60–85%	5	64
Extreme Learning Machine with Radial Basis Function (ELM-R) [19]	62–85%	5	64
Support Vector Machine with Radial Basis Function Kernel (SVM-R) [19]	50–55%	5	64
Linear Discriminant Analysis (LDA) [19]	55–80%	5	64
SVM [17]	22.23%	15	6
Random Forest [17]	23.08%	15	6
rLDA [17]	25.82%	15	6

**Table 2 sensors-21-06503-t002:** Classifiers with DL used for imagined vowel recognition (/a/,/e/,/i/,/o/,/u/).

DL Architecture	Accuracy	Subjects	Electrodes
Deep Belief Networks (DBN) [40]	80%	6	19
Deep Belief Networks (DBN) [18]	87.96%	3	32
Recurrent Neural Networks (RNN) [40]	70%	6	19
Convolutional Neural Networks (CNN) [41]	32.75%	15	6
Convolutional Neural Networks (CNN) [42]	35.68%	15	6
Shallow CNN [17]	29.62%	15	6
Deep CNN [17]	29.06%	15	6
EEGNet [17]	30.08%	15	6

**Table 3 sensors-21-06503-t003:** CNN model specifications of the CNNeeg1-1 architecture.

	Name	Type	Activations	Learnables	Total Learnables
1	Input 32 × 91 × 1 images with ‘zerocenter’ normalization	Image input	32 × 91 × 1	-	0
2	dropout 25% dropout	Dropout	32 × 91 × 1	-	0
3	conv_1 50 5 × 5 × 1 convolutions with stride 1 × 1 and p…	Convolution	28 × 87 × 50	Weights 5 × 5 × 1 × 50 Bias 1 × 1 × 50	1300
4	BN_1 Batch normalization with 50 channels	Batch Normalization	28 × 87 × 50	Offset 1 × 1 × 50 Scale 1 × 1 × 50	100
5	relu_1 ReLU	ReLU	28 × 87 × 50	-	0
6	pool_1 2 × 2 max pooling with stride 2 × 2 and padding…	Max Pooling	14 × 43 × 50	-	0
7	conv_2 60 11 × 11 × 50 convolutions with stride 1 × 1 an	Convolution	4 × 33 × 60	Weights 11 × 11 × 50 × 50 Bias 1 × 1 × 60	363,060
8	BN_2 Batch normalization with 60 channels	Batch Normalization	4 × 33 × 60	Offset 1 × 1 × 60 Scale 1 × 1 × 60	120
9	relu_2 ReLU	ReLU	4 × 33 × 60	-	0
10	pool_2 2 × 2 max pooling with stride 2 × 2 and padding…	Max Pooling	2 × 16 × 60	-	0
11	BN_3 Batch normalization with 60 channels	Batch Normalization	2 × 16 × 60	Offset 1 × 1 × 60 Scale 1 × 1 × 60	120
12	fc1 60 fully connected layer	Fully Connected	1 × 1 × 60	Weights 60×1920 Bias 60×1	115,260
13	fc2 2 fully connected layer	Fully Connected	1 × 1 × 2	Weights 2 × 60 Bias 2 × 1	122
14	softmax softmax	softmax	1 × 1 × 2	-	0
15	classOutput crossentropyex	Classification Output	-	-	0

**Table 4 sensors-21-06503-t004:** Average accuracy of the reference algorithms Shallow CNN, EEGNet, and the new algorithm CNNeeg1-1 for the classification of imagined vowel tasks for intra-subject and inter-subject training using BD1 database.

	Shallow CNN (BD1)		EEGNet (BD1)		CNNeeg1-1 (BD1)	
Model Training	Intra	Inter	Intra	Inter	Intra	Inter
Mean	0.3171	0.2587	0.3506	0.3531	0.6562	0.5008
SD	0.0114	0.0157	0.0133	0.2774	0.0123	0.0133

**Table 5 sensors-21-06503-t005:** Average accuracy of the reference algorithms Shallow CNN, EEGNet, and the new CNNeeg1-1 algorithm in imagined vowel classification tasks for intra-subject and inter-subject training using BD2 database.

	Shallow CNN (BD2)		EEGNet (BD2)		CNNeeg1-1 (BD2)	
Model Training	Intra	Inter	Intra	Inter	Intra	Inter
Mean	0.5371	0.2475	0.7068	0.4578	0.8566	0.6276
SD	0.0606	0.0245	0.0396	0.0433	0.0446	0.0644

## Data Availability

Database BD2 is available online at: https://github.com/carlos-sarmientov/DATABASE-IMAGINED-VOWELS-1, accessed on 4 August 2021.

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
