# Peer review of "Recognition of EEG Signals from Imagined Vowels Using Deep Learning Methods"

_sensors, 2021, doi:10.3390/s21196503_

Round 1

Reviewer 1 Report

Abstract

  • maybe too long; for example, the last sentence is surplus and can be omitted

Introduction

  • the introduction contains let’s say regular paper intro and related work – it would be beneficial for readers to split this section into (shorter) introduction where link with prior work and contribution are clearly pointed out, and into section which contains related work that presents the approaches in more details (and of course, point out their advantages and disadvantages)

Materials and methods

  • blurry images
  • mixing proposed approach and database description (see e.g. 296-298)
  • Figure 5 is not necessary, complete information about CNN structure can be found in Table 1
  • authors give lot of attention to two other models – this should be rather moved into related work or somehow separate from the proposed approach; generally speaking, it is difficult to follow this part of the manuscript due to the language style

Results

  • again, the datasets are described which is not necessary
  • the results should be reported in more concise manner, e.g. take a look at ¸Figure 7 and particularly Figure 8 – e.g. use box plot or similar plot to show the statistics of the obtained results
  • the obtained results are significantly better than the method presented in [25]. However, to prove effectiveness the authors should compare results to the methods like Recurrent Neural Networks (RNN) and Deep Belief Networks (DBN) [39]

Discussion

  • it is not at satisfying level; there is not rationale why the proposed architecture achieves such results

Figures

  •  most of the figures are blurry, are too small etc. à they should be reconsidered (e.g. to have less information but is clearly visible, e.g. Fig. 3, 8…)

Language and style

  • proofreading needed
  • style should be reconsidered too, in some parts manuscript is difficult to follow
  • lot of errors, see for e.g. line 29: c3use à cause, line 259 and so on
  • repeating sentences, eg. line 222
  • unclear language, eg. line 75, 283, 292(?)
  • surplus information (e.g. well known ReLU in Eq (1))
  • formatting is changed in manuscript at several places
  • It appears that authors didn’t use proper template; I cannot find the author contributions, data availability, conflict of interest, and so on which should be written at the very end of the paper. The consents of participants that were involved in BD2 creation should be clearly stated.
  • references not properly written, just to mention [25]

Author Response

We would like to thank the reviewer for their comments. We addressed them in the following way:

  • The Results and Conclusions sections were reorganized to show the results and their relationship with the conclusions more clearly.
  • The indicated sentence in the Abstract was removed.
  • The introduction was split in two, as indicated, in the sections: Introduction and Related Work.
  • The blurry images were replaced with new ones in the *.wmf (Windows Metafile) format to avoid blurring and to allow zooming in without losing sharpness.
  • The description of the proposed architecture is now in section 3.2.1. All content describing the databases in this section was removed.
  • The old Figure 5 was eliminated, as requested.
  • The description of the Shallow CNN and EEGNet architectures was moved to Appendices A and B, respectively.
  • The description of the databases at the start of the Results section was removed.
  • Figures 7, 8, 15, and 16 are now Figures 5, 6, 13, and 14. They were redrawn to show the data more clearly.
  • An explanation about why the proposed architecture offers good results is presented at the end of section 3, lines 326 to 348.
  • We did a proofread of the whole document (including References) looking for spelling, orthography, and grammatical errors and to improve the writing. As for the missing sections at the end (Author contributions, data availability, conflicts of interest, etc), they are included in the new manuscript.
  • About the possible comparison with RNN and BDN, we thank the reviewer for this suggestion, and we will make sure to include these methods in our future work.

We hope to have addressed all your requests properly.

Luis Carlos Sarmiento Ph.D.

Professor UPN

Corresponding Author.

Reviewer 2 Report

see attached file.

Author Response

We’d like to thank the reviewer for their comments. We addressed them in the following way:

  • The paragraph in lines 92 to 96 was rewritten, it is now in lines 107 to 110.
  • The description of the neuroheadset is in lines 177 to 184 and Figure 1.
  • All section headers were reviewed and changed when necessary.
  • The explanation for the proposed architecture can be found at the end of section 3, lines 326 to 348.
  • The description of the EEGNet architecture has been moved to Appendix B, it now includes the explanation for the value of alpha in lines 792 to 794.
  • In the preprocessing of the signals, we used APIT MEMD to decompose them and used only IMF1 and IMF2 for the algorithm. This process reduces the effects of the alpha signals.
  • More detailed explanations for the training processes are provided at the beginning of sections 4.1 and 4.3.
  • The choosing of the hyperparameter values was explained in lines 286 to 288.

We hope to have addressed all your requests properly.

Luis Carlos Sarmiento Ph.D.

Professor UPN

Corresponding Author.

Reviewer 3 Report

Dear authors. I enjoyed reading your manuscript. However, here are some comments for improving it:

  • I would state clearly what are the differences between the two datasets which are used in the study (in the abstract and in the materials). Are the data collected in the same conditions? What are the differences/similarities?
  • Would it be possible to release the code of the study as well in a Github repository?
  • Why accuracy is chosen as a performance metrics? Would it be possible to add other metrics such as precision, sensitivity or specificity?
  • An acronyms section would be of great use.
  • In the discussion section, it would be good to have a comparison table with similar approaches of the literature and the author's approach in order to see the improvement.

Author Response

We’d like to thank the reviewer for their comments. We addressed them in the following way:

  • The characteristics of the databases are explained in sections 3.1.1 and 3.1.2
  • About the availability of the software, only database BD2 is available online at the moment.
  • Other metrics like the ones mentioned by the reviewer could be applied here too. We decided to use accuracy since it is the most used in research in BCI, allowing for easier comparison with other works.
  • An acronyms section has been added in Appendix C
  • The information about previous approaches, presented in section 2, has been reorganized into two tables to make this comparison easier.

We hope to have addressed all your requests properly.

Luis Carlos Sarmiento Ph.D.

Professor UPN

Corresponding Author.

Round 2

Reviewer 2 Report

The authors address my comments. The manuscript is ready for publication.

Reviewer 3 Report

Thank you. All my comments and suggestions have been addressed.